# Bacterial Community Drives the Carbon Source Degradation during the Composting of *Cinnamomum camphora* Leaf Industrial Extracted Residues

**Hanchang Zhou [1,2], Lan Di [1,2], Xiaoju Hua [2], Tao Deng [2] and Xiaodong Wang [2,*]**

[1] Nanchang Urban Ecosystem Monitoring Station, Jiangxi Academy of Forestry, Nanchang 100085, China
[2] Institute of Industrial Forestry, Jiangxi Academy of Forestry, Nanchang 100085, China
[*] Correspondence: xiaodongw2005@163.com

**Abstract:** The increasing production of industrial aromatic plant residues (IAPRs) are potentially environmental risky, and composting is a promising solution to resolve the coming IAPR problems. Carbon source degradation is a basic but important field in compost research; however, we still lack a clear understanding of carbon source degradation and the corresponding relationship to microbial community variation during IAPR composting, which hampers the improvement of IAPR composting efficiency and the promotion of this technology. In this study, samples were chosen on the first day, the 10th day, the 20th day, and the last day during the composting of *Cinnamomum camphora* leaf IAPRs, and the microbial community composition, main carbon source composition, and several enzyme activities were measured accordingly. The results showed that during composting, the hemicellulose had the highest reduction (200 g kg$^{-1}$), followed by cellulose (143 g kg$^{-1}$), lignin (15.5 g kg$^{-1}$), starch (5.48 g kg$^{-1}$), and soluble sugar (0.56 g kg$^{-1}$), which supported that hemicellulose and cellulose were the main carbon source to microbes during composting. The relative abundance of the main bacterial phylum *Firmicute* decreased from 85.1% to 40.3% while *Actinobactreia* increased from 14.4% to 36.7%, and the relative abundance of main fungal class *Eurotiomycetes* decreased from 60.9% to 19.6% while *Sordariomycetes* increased from 16.9% to 69.7%. Though principal coordinates analysis found that both bacterial and fungal community composition significantly varied during composting ($p < 0.05$), structure equation modeling (SEM) supported that bacterial composition rather than fungal counterpart was more responsible for the change in carbon source composition, as the standard total effects offered by bacterial composition ($-0.768$) was about five times the fungal composition ($-0.144$). Enzyme2 (comprised of xylanase, laccase, cellulase and manganese peroxidase) provided $-0.801$ standard total effects to carbon source composition, while Enzyme1 (comprised of lignin peroxidase and polyphenol oxidase) had only 0.172. Furthermore, xylanase and laccase were the only two enzymes appeared in co-occurrence network, clustered with nearly all the carbon sources concerned (except starch) in module-II. Xylanase, hemicellulose, and cellulose were linked to higher numbers of OTUs, more than laccase and other carbon sources. In addition, there were 11 BOTUs but only 1 FOTUs directly interacted to xylanase, hemicellulose, and cellulose simultaneously, three of them were *Limnochordaceae* and two were *Savagea*, which highlighted the potential core function in lignocellulose degradation provided by bacterial members, especially *Limnochordaceae* and *Savagea*. Thus, the results supported that during composting of *Cinnamomum camphora* leaf IAPRs, the degradation of dominate carbon sources, hemicellulose and cellulose, was mainly driven by bacterial community rather than fungal community. In addition, the bacterial originated xylanase and laccase played potentially core roles in the functional modules. This research clearly investigated the microbial dynamics of carbon source degradation during the composting of *Cinnamomum camphora* leaf IAPRs, and offers valuable information about and new insight into future IAPRs waste treatment.

**Keywords:** compost; carbon source; microbial community; *Cinnamomum camphora*; industrial extracted residues

## 1. Introduction

Aromatic plants provide varieties of high value secondary metabolites that play irreplaceable roles in the pharmaceutical [1], cosmetic [2] and food additive industries [3]. In addition, as the upsurging raw materials demand, the cultivation area, the yielding of aromatic plants, and the industrial extraction also expand all over the world [1–3]. Yet, most commercially targeted secondary metabolites only use a low proportion of plant biomass, and over 90% of the biomass are left as industrial aromatic plant residues (IAPRs) after industrial extraction [4,5]. For example, *Cinnamomum camphora*, a common aromatic plant that is widely distributed, had as high as $6.67 \times 10^4$ hectares of cultivation area just in China since 2019 [6]. Nearly 0.98 t dry mass residues are produced for every 1 t dry mass plant, which indicated a huge organic waste production from *Cinnamomum camphora* industrial extraction [6]. Such a prominent organic waste input by IAPRs would cause several environmental problems, such as eutrophication and odorization [7]. Moreover, different to common organic waste (like straws or manures) from agriculture and husbandry, IAPRs are possibly mixed with ecotoxic organic solvents, such as ether and acetone [4–6], which diffuse the soil and atmosphere and cause potential alterations to ecosystem functions [8]. The secondary metabolite remaining in the residues have antibiotics functions, which may lead to other unexpected negative ecological effects, such as the enrichment of antibiotic resistance genes [4–6]. We urgently need a sustainable measure to consume the increasingly emergent IARPs. Unfortunately, the treatment for *Cinnamomum camphora* industrial extracted residues, as well as other IAPRs, are seldom reported.

Composting, a traditional organic waste treatment, is widely adopted as a recycling method in the agriculture, husbandry and municipal fields, and offers a new sight for resolving the expanding IAPRs problems [9]. Carbon source decomposition is an area of great interest in composting research, which is directly related to the quality of the compost products and the release of greenhouse gases, such as $CO_2$ and $CH_4$ during the composting process [10]. During composting, the recalcitrant organic polymers in the raw materials are degraded into labile small organic molecules by microbes, then part of the small organic molecules are assimilated or respired for the growing, metabolite and reproduction of microbes [10]. Thus, the carbon content of composted material decreases gradually, while the content of nitrogen, phosphorus and potassium increase accordingly, which enhances the fertility of compost materials and turns the raw compost materials into potential fertilizers [9,10]. The transformation and degradation of recalcitrant organic polymers in the raw materials are reported to be highly connected to the microbial community composition and the chemical structure of raw materials [11–13], while these two factors are always mutually influenced. For example, inoculating certain species of active microbes could accelerate the carbon source transformation, and the enhanced carbon turnover rate could in turn promote the growth and activity of microbial communities and increase the compost efficiency [14,15]. In addition, the less energetic raw materials, or lower nitrogen content materials favors K-strategic microbes and reduced the competence of r-strategic microbes, and the energy conservative microbial community in turn decrease the carbon source transformation rate [16]. As the compositional and functional dynamics of microbial communities and the carbon source transformation and degradation are highly correlated, a clear understanding of the relationship between carbon source transformation and microbial community dynamics would offer substantial improvements to composting processes as well as the quality of compost products [16].

The plant originated carbon sources in common compost materials are generally characterized as high molecular organic polymers, such as cellulose, hemicellulose, lignin and starch [10]. In addition, the capability of microbial community to use different organic polymers depends on the activity of varieties enzymes released by microbial community members [11–13,16]. Fungi were widely reported able to release cellulase and xylanase that breakdown the covalent bonds among the units that form cellulose and hemicellulose [17]. In addition, the degradation of lignin, a highly aromatized polymer, were engaged to

series enzymes including laccase, lignin peroxidase (LiP), manganese peroxidase (MnP), and polyphenol oxidase, which were reported to be mainly released by soil fungi [18,19]. Several papers also highlighted the contribution of bacterial enzymes in compost material transformation, as most bacterial enzymes were more resilient to environment variations, possessed more stable activities, and had higher industrial application value than fungal counterparts [10,20]. Microbial communities release different enzymes according to the composition and accessibility of carbon sources [10,19], which leads to predicting microbial dynamics and the corresponding relationships to carbon sources being a hard question to answer [10]. In addition, this question is made more difficult as concerns IAPRs compost, because compared to traditional agricultural and husbandry compost materials, IAPRs suffer several processes, such as high temperature boiling and organic solvent extraction [6], and IAPRs contain varieties of secondary metabolite residues that possess antibiotic activities [1,6]. The boiling might break the combination of lignin, cellulose and hemicellulose, expose the enclosed cellulose and hemicellulose and enlarge the bio-accessibility of cellulose and hemicellulose [21], which in turn favors r-strategic microbes, such as bacteria rather than fungi [16]. While boiling might also dilute the content of original small organic molecules as they are always hydrophilic, these reduce the labile carbon source availability to r-strategic microbes [6,21]. In addition, the antibiotic secondary metabolite left in IAPRs are possibly unfriendly to certain types of microbial community members, disrupting their metabolisms and carbon use, which alters the bio-accessibility of carbon sources indirectly [1]. Industrial extraction lead potential change on the physical and chemical properties of natural carbon sources, which is a sign of the change on carbon source bio-accessibility, in turn influences on the microbial community composition and the enzyme released during composting [6].

To elucidate the detailed microbial dynamic of carbon source degradation during IAPRs composting, here we sampled materials at four different stages during the composting of *Cinnamomum camphora* leaf IAPRs, and measured the variation of carbon source composition, microbial community composition, and the activity of several carbon source degradation related enzymes. Considering the generally stronger capability of fungi in decomposing recalcitrant organic carbons in the plant material composts, we hypothesize that during IAPRs compost, fungal community composition would better explain the carbon source composition change, and the carbon metabolism related enzymes would be linked more to fungal members rather than bacterial members.

## 2. Materials and Methods

### 2.1. Study Sites and Sampling

The composting site was at the Man-De-Seng Agricultural Development Limited Company (Ji'an City, Jiangxi Province, E116°07′67.4″ and N28°34′95.42″). The compost materials were an even mixture of *Cinnamomum camphora* leaf industrial extracted residues (offered by Gaosheng Hi-Tech Limited Company, crushed to size less than 3 cm) and fresh quail manure (offered by an unnamed quail breeding ground), with a ratio of 4:1 (mass/mass). The *Cinnamomum camphora* leaves were boiled under 115 °C 0.5 MPa for 65 min during industrial extraction. The extra decomposer additives were added with a quantity about 0.15% mass of total compost materials, and the additives were purchased from Jiwei (Hebei) biotech. The compost pile was about 3 m × 2 m × 1.6 m and three piles were set as duplications. The materials were remixed every 10 days. We sampled at 0th, 10th, 20th, and 30th day of composting, and marked the samples as group A, B, C and D, respectively. During each sampling, 10 subsamples (each around 500 g) were randomly selected, followed by passing through 2 mm sieves, then about 1000 g mixed materials were chosen as one sample. Fifty grams were stored at −80 °C for a DNA sequencing test, and others were transported to the lab for measuring of basic physicochemical properties as soon as possible. Materials were kept about 55% gravimetric moisture during the whole composting, by daily measurement and artificial watering.

*2.2. Determination of Basic Physicochemical Properties and Enzyme Activities*

The water content of compost materials was measured by gravimetric methods. pH was determined in a compost-water mixture (1:5 ratio of mass/volume) by a pH meter (FE20-FiveEasyTM pH, MettlerToledo, Berlin, Germany). Total carbon and total nitrogen of compost materials were measured by element analyzer (Vario MACRO cube, Elementar Inc., Germany). Total phosphorus was measured by the Sommers-Nelson method [22]. Total potassium was measured by the flame-spectrometric method [23]. The content of cellulose, hemicellulose, lignin, starch, soluble sugar and protein, and the activity of xylanase, cellulase, laccase, MnP, LiP and polyphenol oxidase were all measured according to the instruction of corresponding kit (Comin Biotechnology Co., Ltd., Suzhou, China). The conductivity was measured by a conductivity meter with probe (Leici, DDS-307A, INESA, Shanghai, China).

*2.3. The DNA Sequencing and Bioinformatic Analysis*

Microbial DNA was extracted using FastDNATM SPIN kit (MP Biomedicals, Los Angeles, CA, USA), and the DNA concentration and quality were determined by a Nano-100 NanoDrop spectrophotometer. Primers for 16S rRNA gene amplification targeted the V3-V4 hypervariable region and included 338F 5′-ACTCCTACGGGAGGCAGCAG-3′ and 806R 5′-GGACTACHVGGGTWTCTAAT-3′ [24]. The primers for ITS1 region amplification included 1F 5′-CTTGGTCATTTAGAGGAAGTAA-3′ and 2R 5′-GCTGCGTTCTTCATCGATGC-3′ [25]. PCR was conducted according to previous research and the amplicons were pooled in equimolar ratios, sequenced in paired-end form on the Illumina Nova6000 platform (Majorbio Company, Shanghai, China) [26]. Majorbio Cloud Platform (www.majorbio.com (accessed on 10 September 2022)) was adopted for the treatment of the bioinformatics scripts, abd sequences were merged with a minimum overlap length of 20 bp into full-length sequences by FLASH after removing barcodes and primers [27,28]. After removing the chimeras with UPARSE, the sequences of more than 97% similarity were clustered into operational taxonomic units (OTUs) (Edgar, 2013). The Sliva-138 database and UNITE-8.0 database were chosen to annotate bacterial OTUs and fungal OTUs, respectively. Reads number of 28,885 (bacterial) and 37,645 (fungal) sequences were randomly selected from each sample to form an evenly resampled OTU table for further analysis.

Network analyses were conducted according to the molecular ecological network analysis pipeline (MENA, http://ieg4.rccc.ou.edu/MENA/login.cgi, accessed on 10 September 2022) protocols [29]. The BOTUs (bacterial OTUs) and FOTUs (fungal OTUs) were combined to construct co-occurrence network. The construction parameters were set as majority = 9, missing_fill = fill_paired (0.0100), logarithm = n, similarity = spearman2, cutoff threshold was 0.85. Calculations of richness, Shannon index, principal coordinates analysis (PCoA) based on Bray-Curtis distances at OTU-level, analysis of Similarities (ANOISM), linear discriminant analysis effect size (Lefse analysis) were conducted with the online platform https://cloud.majorbio.com, accessed on 27 January 2023, under R base [30].

*2.4. The Statistics*

The structure equation modeling (SEM) was conducted with Amos v18.0 and SPSS v24, the extractions were conducted according to Pearson correlation metrics, and divided into different groups for modeling construction [31]. The PC1 of bacterial and fungal PCoA results were used as the composition parameters of bacterial community and fungal community, respectively [26]. The significance of differences among different groups was checked using one-way ANOVA (Tukey's HSD test) on SPSS v24. The pictures were visualized by Origin.v16.0, and co-occurrence network was visualized by Cytoscape.v3.3.0.

## 3. Results

The moisture of initial compost materials was about $54.14 \pm 2.79\%$, pH was $7.05 \pm 0.06$, total carbon was $9.78 \pm 0.82\%$, total nitrogen was $2.17 \pm 0.16\%$, total phosphorus was

3.09 ± 0.02 mg g$^{-1}$, total potassium was 18.79 ± 1.47 mg g$^{-1}$. In addition, we will show our main results in the following subsections.

### 3.1. The Variation on Carbon Source and Enzymes

The content of cellulose, hemicellulose, lignin, starch, and soluble sugar showed decreasing trends during composting while protein showed increasing one (Figure 1A,B,E,F,I,J), which indicated the biosynthesis of microbes with consumption of carbon sources. The hemicellulose had highest reduction from 222 ± 17.1 g kg$^{-1}$ (A) to 22.8 ± 1.17 g kg$^{-1}$ (D) and nearly 90% were degraded (Figure 1E). Cellulose decreased around 60%, from 258 ± 6.16 g kg$^{-1}$ (A) to 115 ± 4.83 g kg$^{-1}$ (D) (Figure 1A). Compared to other carbon sources, hemicellulose and cellulose were significantly correlated with protein ($p < 0.05$), and the correlation coefficients were, respectively, −0.594 and −0.559 (Figure 1M). Lignin also showed nearly 50% decrease, from 46.2 ± 4.79 g kg$^{-1}$ (A) to 30.7 ± 1.89 g kg$^{-1}$ (D), both the quantity and ratio of reduction were less than hemicellulose and cellulose (Figure 1I). Starch was around 55% reduced during compost, but it was the least polymer at initial stage, only 9.48 ± 0.48 g kg$^{-1}$ (A), which was only around 5% and 4% of cellulose and hemicellulose, respectively (Figure 1B). Soluble sugar weakly reduced from 2.98 ± 0.16 g kg$^{-1}$ (A) to 2.61 ± 0.11 g kg$^{-1}$ (D) (Figure 1F), and was significantly correlated with four polymer carbon sources ($p < 0.05$) (Figure 1M).

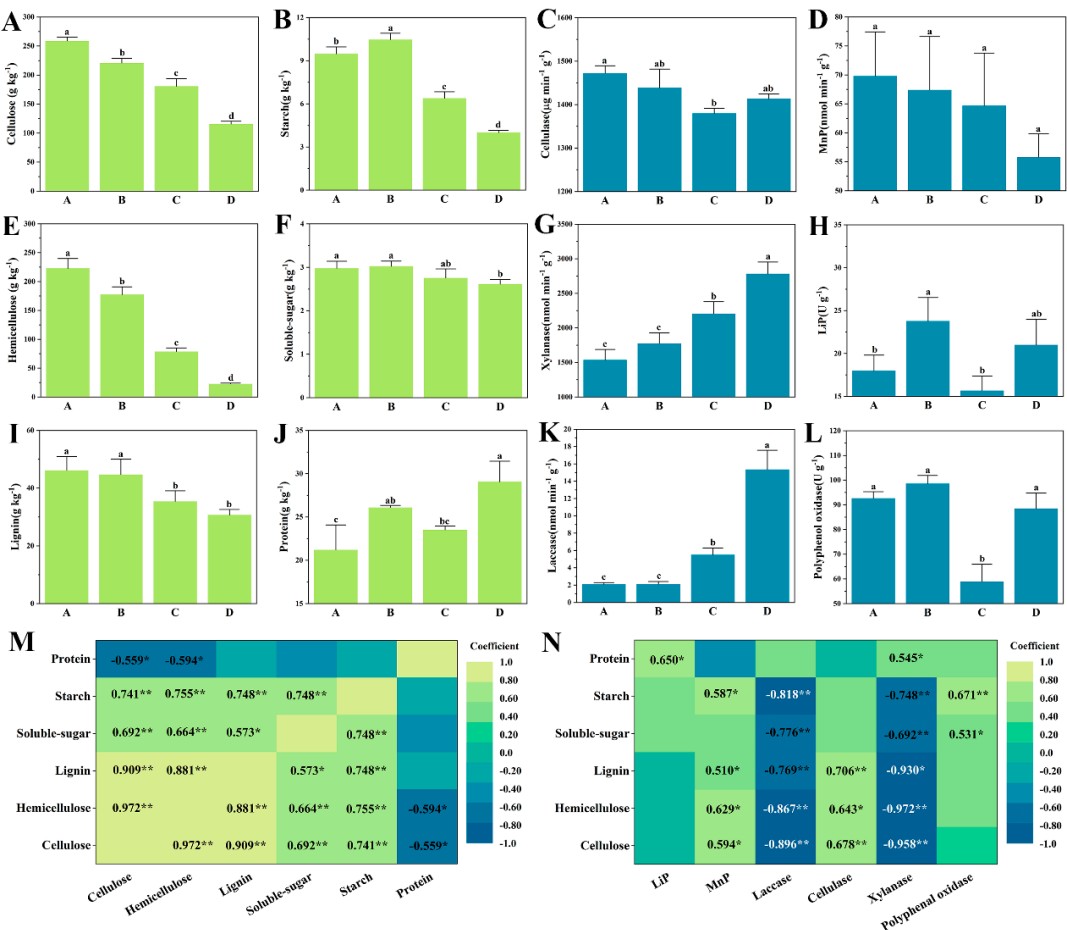

**Figure 1.** The variation of carbon sources and enzyme activities. (**A–L**) depicts the variation of carbon sources and enzyme activities during four stages, different letters a, b, c and d indicate significant differences at $p < 0.05$ level (Turkey's HSD). (**M,N**) depict the Spearman correlation heatmaps of carbon sources and enzyme activities, * indicates significant correlated at $p < 0.05$ level and ** $p < 0.01$ level, blue indicates negatively correlated while yellow positively. The LiP and MnP are abbreviations for lignin peroxidase and manganese peroxidase, respectively.

The activity of xylanase gradually increased nearly 0.8 times, from $1537 \pm 150$ nmol $min^{-1} g^{-1}$ (A) to $2782 \pm 171$ nmol $min^{-1} g^{-1}$ (D) (Figure 1G), and was significantly correlated with other carbon sources and protein (Figure 1N). The cellulase activity decreased weakly, which was lowest at stage B ($1380 \pm 11.6$ µg $min^{-1} g^{-1}$) and about 10% lower than A ($1472 \pm 16.7$ µg $min^{-1} g^{-1}$) (Figure 1C). Cellulase was significantly correlated with the content of cellulose, hemicellulose, and lignin, which indicated a co-metabolism among those high molecular polymers (Figure 1N). Laccase activity increased after stage B, and reached $15.3 \pm 2.22$ nmol $min^{-1} g^{-1}$ (D), which was nearly 6.5 times higher than A and B (Figure 1K). However, MnP decreased from $70.0 \pm 7.61$ nmol $min^{-1} g^{-1}$ (A) to $55.8 \pm 4.00$ nmol $min^{-1} g^{-1}$ (D), though no significant differences were found among the four stages (Figure 1D). The polyphenol oxidase activity was lowest in C ($59.0 \pm 7.00$ U $g^{-1}$) that was significantly lower than the other three stages ($p < 0.05$) (Figure 1L). LiP activity was also lowest at C ($15.7 \pm 1.70$ U $g^{-1}$) and highest at B ($23.8 \pm 2.73$ U $g^{-1}$) (Figure 1H). Laccase activity was significantly correlated with all carbon sources ($p < 0.05$), MnP was also significantly correlated with nearly all carbon sources but soluble sugar ($p < 0.05$), polyphenol oxidase was significantly correlated with starch and soluble sugar ($p < 0.05$), and LiP was only correlated with protein ($p < 0.05$) (Figure 1N). These indicated that laccase might contribute mainly for lignin degradation, and those lignin degradation related enzymes were functionally preferential.

### 3.2. Microbial Community Composition Variation

At stage A, bacterial Shannon index was $3.41 \pm 0.02$, which was significantly lower than B ($3.98 \pm 0.08$) ($p < 0.05$), while C and D had a medium Shannon index and no significant differences were found between them ($p > 0.05$) (Figure 2A). Bacterial Chao1 index was highest at C ($601 \pm 173$) and lowest at A ($419 \pm 94$), but there were no significant differences among each stages ($p > 0.05$) (Figure 2B). Bacterial community was mainly occupied by *Firmicutes* and *Actinobacteriota* at stage A, they had relative abundances (RA) of 85.1% and 14.4%, respectively (Figure 2C). With the ongoing of composting, RA of *Actinobacteriota* gradually increased to 36.7%, while *Firmicutes* decreased to 40.3%. In addition, the RA of *Proteobacteria*, *Bacteroidota*, *Myxococcota*, *Gemmatimonadota,* and others also showed increasing trends. For instances, the *Myxococcota* appeared only at stage D (8.07% RA), *Gemmatimonadota* and others also possessed considerable RA at D (5.41% and 2.71%, respectively) (Figure 2C). PCoA analysis found that the bacterial community composition varied significantly during composting (ANOISM $p < 0.05$), but C and D were mostly overlapped (Figure 2D). Lefse analysis found that stages B and C had few BOTUs while stage A and stage D had respectively 10 and 6 BOTUs, which indicated remarkable OTU level differences at the initial and end of composting, rather than medium stages (Figure 2E).

Fungal Shannon index significantly decreased at stage D ($1.94 \pm 0.83$) ($p < 0.05$) (Figure 2F). In addition, Chao1 index was also lowest at D ($158 \pm 54$), but there were no significant differences among stages ($p > 0.05$) (Figure 2G). The variation of fungal community composition was characterized as the trade-off between *Eurotiomycetes* and *Sordariomycetes*. Stages A, B, and C = were mainly occupied by *Eurotiomycetes*, which RA ranged between 60.9% (B) and 67.4% (A) (Figure 2H). The *Sordariomycetes* was the second largest group, which RA ranged between 16.9% (A) and 11.0% (C). However, at stage D, RA of *Sordariomycetes* reached as high as 69.7%, which had surpassed *Eurotiomycetes* (RA 19.6%). RA of *Tremellomycetes* showed an increasing trend and reached highest RA at C (11.2%). Stage B possessed unique appearance of *Saccharomycetes* (RA 7.32%), which the RA at other stages were nearly zero (Figure 2H). PCoA analysis reported significant changes in fungal community (ANOISM $p < 0.05$), especially at stage D (Figure 2I). Lefse analysis showed that stages A and B had 7 and 9 remarkable FOTUs, respectively, while C and D had only FOTU517 and FOTU912, respectively (Figure 2J).

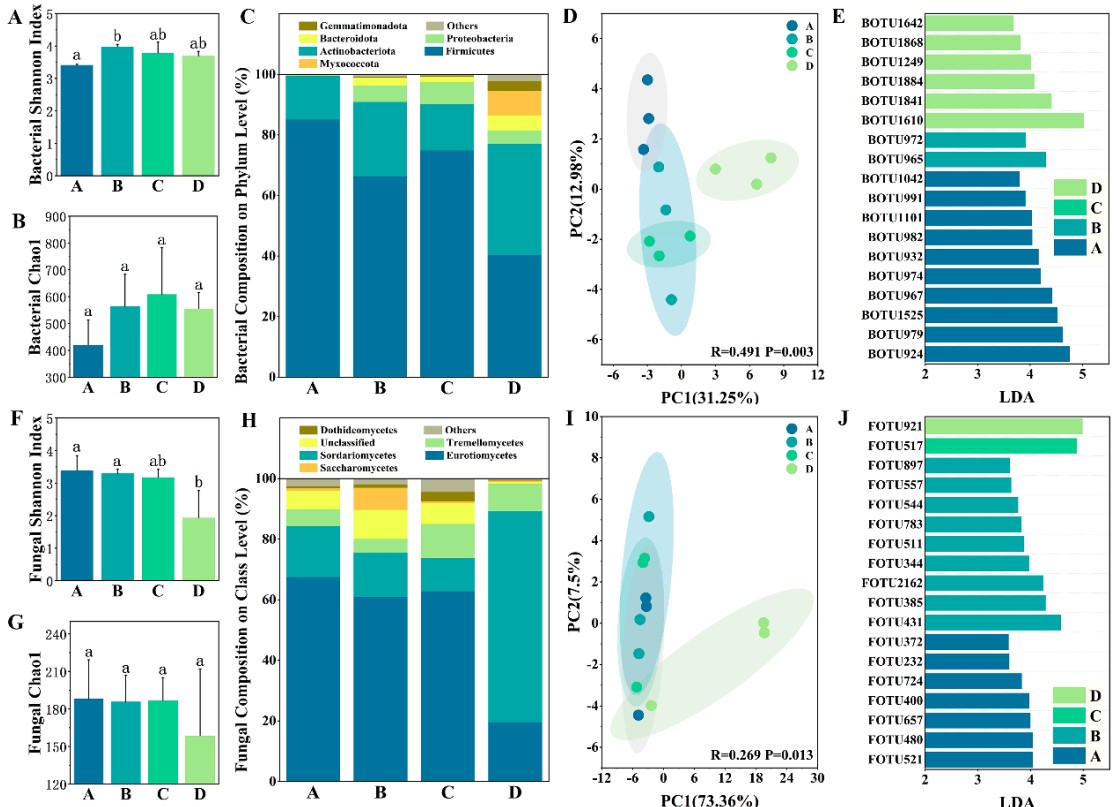

**Figure 2.** Variation on microbial community diversity indexes, composition, and distinctive members. (**A**,**B**,**F**,**G**) depicts the change in Shannon indexes and Chao1 indexes, the letters a and b indicate significant differences at *p* < 0.05 level (Turkey's HSD). (**C**,**H**) depicts the compositional barplot. (**D**,**I**) depicts the PCoA analysis results, the R-value and *p*-value are generated from ANOISM analysis. (**E**,**J**) depicts Lefse analysis results of bacterial and fungal OTUs, respectively. The OTU with LDA-value more than 3.5 are exhibited.

### 3.3. Contribution to the Variation of Carbon Source Composition

According to SEM, Enzyme2 (extracted by xylanase, cellulose, laccase, and MnP) contributes the highest standard total effects −0.801 to carbon source composition, followed by bacterial community composition (−0.768 standard total effects), which were far higher than the magnitude of standard total effects offered by fungal community composition (−0.144) and Enzyme1 (extracted by polyphenol oxidase and LiP, 0.172). Enzyme2 also offered highest standard direct effects to carbon composition (−0.695), which was higher than bacterial community composition (−0.353), but bacterial community exerted higher standard indirect effects (−0.415) than Enzyme2 (−0.141) (Figure 3). These indicated that the bacterial community, rather than the fungal community, is the dominate microbial contributor to carbon composition variation in composting, mainly by indirect effects of influencing Enzyme2, instead of Enzyme1.

### 3.4. Network Analysis

There were three main modules in the network, and module-II was carbon source transformation and degradation related, since most carbon sources were contained in module-II, except starch. The whole network contained 179 nodes and 795 links, and nearly 80% of links were positive and mainly appeared in module-I, which indicated the frequent co-metabolite and cooperation among microbes in composting. In addition, the negative links mainly appeared between module-I and module-II, indicating potential competition between two modules. There were only xylanase and laccase located in module-II, highlighted that they play core roles in carbon sources transformation and degradation during composting (Figure 4).

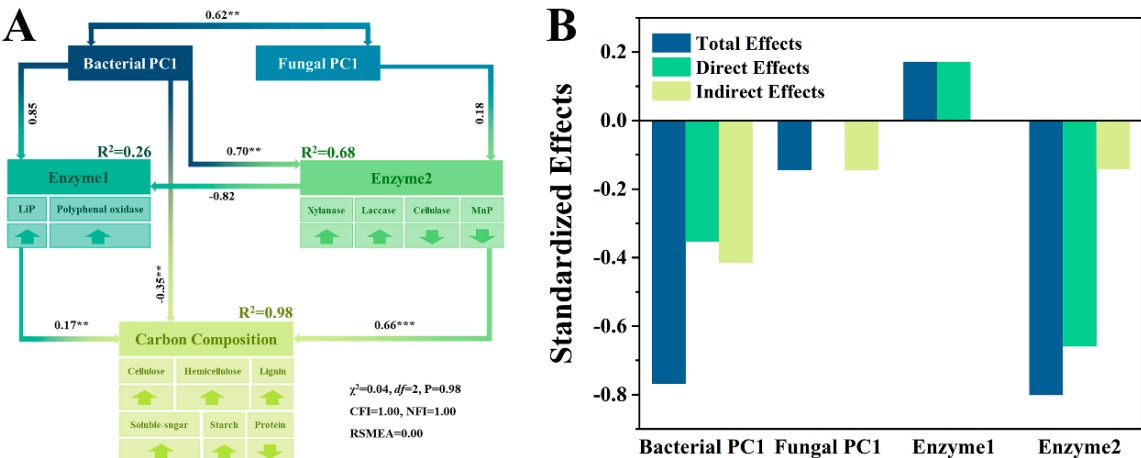

**Figure 3.** Structure equation modeling resolving carbon composition variations. (**A**) depicts the model, the up and down arrows indicate positively and negatively correlated with the extractions, ** and *** indicated significantly related at $p < 0.05$, $p < 0.01$ and $p < 0.001$ level, respectively. (**B**) depict the standardized effects on carbon source composition. The LiP and MnP are abbreviations for lignin peroxidase and manganese peroxidase, respectively.

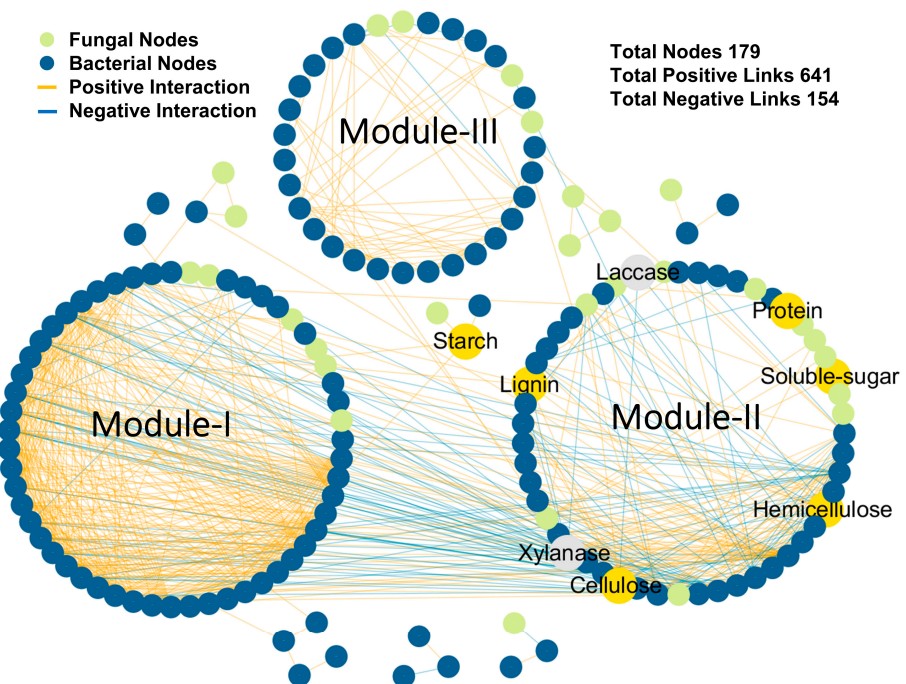

**Figure 4.** The co-occurrence network of microbial community in compositing. The orange and blue line indicate positive and negative interaction, respectively. The dark blue, green, yellow, and gray dot indicate bacterial member, fungal member, carbon source and enzyme, respectively.

For further analysis on the detailed OTU identification that directly linked to enzymes and carbon sources, we found xylanase was negatively linked to 17 OTUs and positively linked to 8 OTUs. In addition, hemicellulose had eight negatively linked OTUs and six positively linked OTUs, cellulose had six negatively linked and seven positively linked OTUs. There were 12 OTUs directly interacting with xylanase, hemicellulose, and cellulose simultaneously, three of them were Limnochordaceae and two of them were Savagea. However, laccase was only directly linked to BOTU967 and BOTU991. Both lignin and starch were linked to 4 OTUs but soluble sugar was only linked to FOTU400 and FOTU425, and protein was only linked to BOTU1570. Interestingly, BOTU991 (Genus Savagea) was

linked to xylanase, laccase, hemicellulose, cellulose and lignin, which indicated a full metabolite capability for the degradation of organic polymers (Table 1).

**Table 1.** The OTUs directly interact to the enzymes and carbon sources in the network.

| OTU ID | Genus | Xylanase | Cellulose | Hemicellulose | Laccase | Lignin | Starch | Soluble Sugar | Protein |
|---|---|---|---|---|---|---|---|---|---|
| BOTU1023 | *Lactobacillus* | − | | | | + | | | |
| BOTU1051 | *Corynebacterium* | − | | | | | | | |
| BOTU1114 | *Brevibacterium* | | | | | | + | | |
| BOTU1137 | *Aerosphaera* | − | | − | | | | | |
| BOTU1145 | *Enterococcus* | | | | | | + | | |
| BOTU1181 | *Carnobacteriaceae* | | | | | + | | | |
| BOTU1236 | *Limnochordaceae* | | | − | | | | | |
| BOTU1249 | *Limnochordaceae* | + | − | − | | | | | |
| BOTU1570 | *Bacillaceae* | | | | | | | | − |
| BOTU1610 | *Thermobifida* | + | − | − | | | | | |
| BOTU1613 | *Ammoniphilus* | + | − | − | | | | | |
| BOTU1642 | *Bacillaceae* | + | − | − | | | | | |
| BOTU1663 | *Paenibacillaceae* | + | | | | | | | |
| BOTU1715 | *Limnochordaceae* | + | − | − | | | | | |
| BOTU1769 | *Limnochordaceae* | + | − | − | | | | | |
| BOTU1790 | *Novibacillus* | + | | | | | | | |
| BOTU890 | *Jeotgalibaca* | − | | | | | | | |
| BOTU898 | *Brevibacterium* | − | | | | | | | |
| BOTU924 | *Atopostipes* | − | + | + | | + | | | |
| BOTU932 | *Sporosarcina* | − | + | + | | | | | |
| BOTU959 | *Staphylococcus* | − | | | | | | | |
| BOTU963 | *Facklamia* | − | | | | | | | |
| BOTU965 | *Corynebacterium* | − | | | | | | | |
| BOTU967 | *Pseudogracilibacillus* | | | | − | | | | |
| BOTU972 | *Gallicola* | − | | | | | | + | |
| BOTU974 | *Gallicola* | − | + | + | | | | | |
| BOTU979 | *Savagea* | − | + | + | | | | | |
| BOTU991 | *Savagea* | − | + | + | − | + | | | |
| FOTU400 | *Aspergillus* | − | + | | | | | + | |
| FOTU425 | *Scopulariopsis* | | | | | | | + | |
| FOTU511 | *Pestalotiopsis* | | | | | | + | | |
| FOTU521 | *Aspergillus* | − | + | + | | | | | |
| FOTU727 | *Penicillium* | − | | | | | | | |

Note: + and − indicate positive and negative interaction, respectively.

## 4. Discussion

### 4.1. The Variation of Carbon Sources and Enzyme Activities during Composting

In this research, hemicellulose and cellulose were the main carbon sources for microbes, as their quantities were about 5~20 times higher than other carbon sources and they had the highest decrement proportion, especially hemicellulose (Figure 1). Cellulose, hemicellulose and lignin are reported to be the main carbon sources in plant material compost, and as they are always chemically bonded and physically twisted, their biodegradation is also closely connected [10,32], just as our network analysis results showed that the cellulose, hemicellulose, and lignin were clustered in module-II (Figure 4). However, cellulose, hemicellulose, and lignin are still different in bio-accessibility due to the differences in the chemical unit as well as the differences on the state in which they behave when forming the complex compounds [10,32]. Cellulose consists of highly fibrous assemblies comprised by glucose units, which tend to form as somewhat crystalline, while hemicellulose is less fibrous than cellulose and does not readily form crystalline domains, which make hemicellulose more accessible to enzymes and easier to be degraded by microbes [10,32]. In addition, lignin is a phenolic polymer material instead of polysaccharides, its degradation

needs the complex enzyme systems produced by microbes, while unit carbon in lignin offers less energy than polysaccharide does, which makes lignin less favorable to microbes than cellulose and hemicellulose [10,33]. In this research, hemicellulose and cellulose decreased significantly at the early stages, while lignin significantly reduced from stage C (Figure 1A,E,I), indicating that microbial community also prefers hemicellulose and cellulose during the composting of *Cinnamomum camphora* leaf IAPR.

In this research, xylanase activity increased during composting (Figure 1G), and was significantly correlated with the content of cellulose, hemicellulose, lignin, soluble sugar, starch, and protein (Figure 1N), which indicated that xylanase was the driver to organic polymer degradation during this composting. Xylanase is mainly responsible for hemicellulose degradation [17]. Parts of the hemicellulose links to lignin by covalent bound, and helps to form a better integration of the inherently hydrophilic cellulose with the much less hydrophilic lignin component [34]. The breakdown of intermedium hemicellulose releases the twisted cellulose and lignin, thus make cellulose and lignin more accessible to microbes and promote their degradation [21,34]. Our results showed that xylanase is negatively correlated with hemicellulose, while cellulase is positively correlated with cellulose (Figure 1N), which furtherly indicated that the degradation of hemicellulose accelerated the production and releasing of cellulase, and thus enhanced the use of cellulose. Laccase, MnP, LiP and polyphenol oxidase are responsible for lignin degradation. Laccases are blue copper oxidase enzymes, and soil laccases were released by most fungi and several bacteria [35]. Laccases were reported to cooperate with either MnP or LiP to lead lignin degradation, because the redox potentials of the laccases was not high enough to remove electrons from nonphenolic aromatic parts of lignin [10]. MnP is widely distributed among white-rotters [36], and is capable of oxidizing phenolic lignin units [10]. In this research, laccase showed highest coefficient to lignin, followed by MnP, while LiP and polyphenol oxidase were not significantly correlated with lignin (Figure 1N). Furthermore, both laccase and MnP were significantly correlated with cellulose and hemicellulose (Figure 1N). These indicated that laccase and MnP were more responsible for lignin degradation during this composting.

*4.2. Relationship between Microbial Community and Carbon Source Degradation*

Our research supports that during composting of *Cinnamomum camphora* leaf IAPRs, bacterial community composition rather than fungal community composition plays more important roles in governing carbon source transformation and degradation, mainly through Enzyme2 (xylanase, laccase, cellulase, and MnP) (Figure 3). This is inconsistent with previous agricultural [14] and husbandry compost research that highlighted the fungal community functions in the degradation of organic polymers. Nitrogen was reported as being beneficial to bacterial community activities in composting, and our compost material had a lower C:N ratio, which was a potential reason for why the bacterial community out-competed the fungal community [16]. Moreover, the cellulose and hemicellulose in traditional agricultural and husbandry compost materials are always physically protected by lignin shelter [34,37]. In addition, fungal hyphae can insert into the shelter, release the nutrients protected, which make fungi more adaptive for lignocellulose degradation [10]. Yet, the lignocellulose of IAPRs suffer industrial processes such as long boiling time and organic solvent extraction [6], which somewhat break the lignin shelter and make the protected nutrients = become more accessible, thus enhancing bacterial community competence [10]. Though xylanase and laccase were widely discovered in fungi and were frequently applicated to traditional compost engineering [38], the reports about bacteria originated xylanase and laccase were also rising in recent years. Previous research reported their structure-based semi-rational engineering on bacterial Lac15 that increased the sensitivity of Lac15 to parts of the substrate [39]. A strain of *Raoultella ornithinolytica* RS-1 was reported to be isolated from the termite gut and reported its laccase and LiP activities [40]. The *Bacillusamyloliquefaciens* MN-13, which isolated from caw manure, was also reported to be able to degrade 18% of the total lignin in pure cultivation [41]. Bacterial laccases

can work under wider pH extent and higher temperature condition, and are reported to possess higher industrial application potential [20]. Thus, we suggest developing IAPRs that compost specific bacterial enzymes and inoculants to enhance compost efficiency.

In this research, RA of *Firmicutes* varied most prominently, as the RA decreased from nearly 80% (A) to 40% (D), while *Actinobacteria* increased gradually (Figure 2C). *Firmicutes* were also reported to be the dominant group of municipal waste compost, which the RA ranged between 35% to 50% and followed by *Actinobacteria* [42]. The oxygen level in the compost was considered to be the main cause of the trade-off between those Gram-positive bacteria and the ubiquitous facultative anaerobic bacteria [42]. Moreover, the disappearance of phytotoxicity during the maturation phase favored the increase in *Actinobacteria* RA [43]. Thus, the enhanced *Actinobacteria* RA in this research was a sign for the degradation of secondary metabolite in *Cinnamomum camphora* leaf IAPRs, as the *Cinnamomum camphora* secondary metabolites were widely reported to inhibit the growth of microbes [6]. The RA of *Firmicutes* were also highest in cow manure compost, especially in the middle stage of composting, which supported that *Firmicutes* were relatively thermophilic and likely degraded lignocellulose in the whole composting process [44,45], while *Actinobacteria*, less thermophilic than *Firmicutes*, was reported as being capable of stimulating microbes to produce lignocellulosic hydrolases, increasing the decomposition of organic matters, and enhancing the mineralization of organic nitrogen [46]. *Actinobacteria* also secreted a variety of antibiotics to reduce the growth and regeneration rates of pathogenic microorganisms, promoted the safety and quality of compost products [47].

The RA of *Sordariomycetes* out-competed *Eurotiomycetes* at stage D in our research (Figure 2H). *Sordariomycetes*, saprotrophic fungus, were the main genera in the middle and late compost stage, and played important roles in the circulation of nutrients as well as the degradation of recalcitrant organic matters (like lignin) during plant material waste compost [44,48]. In addition, *Eurotiomycetes* is a monophyletic class of *Ascomycota*, which has high activity of carbohydrate metabolism, especially for cellulose and the small molecular organic acids that generated from the degradation of organic polymers [49]. *Eurotiomycetes* contains several members that have high economic value, such as antibiotic production engaged *Penicillium chrysogenum* and plant material fermentation engaged *Aspergillus oryzae* [49]. *Aspergillus*, coupled with *Mycothermus*, dominated the fungal community at the thermophilic phase of composting, and positively correlated with parameters of carbon and nitrogen transformation, as *Aspergillus* produced a series of thermostable enzymes [50]. As mentioned above, the microbial community preferred using hemicellulose and cellulose as carbon source than lignin, and considered that the *Sordariomycetes* and *Eurotiomycetes* favor lignin and cellulose, respectively. The trade-off between *Sordariomycetes* (Stage D) and *Eurotiomycetes* (Stages A, B, and C) was possibly due to the variation of hemicellulose, cellulose and lignin.

We found that module-II is highly related to carbon source transformation and degradation (Figure 4). The members in same module are always engaged to same ecological functions [29], which indicates the potential and intimate co-metabolism of cellulose, hemicellulose and lignin in composting of *Cinnamomum camphora* leaf IAPRs. Only xylanase and laccase appeared in the co-occurrence network (Figure 4), which strengthened what the metabolism hub functions of xylanase and laccase offered to the degradation of organic polymers. The laccase directly linked to 2 BOTUs and lignin directly linked to 4 BOTUs, which indirectly supported the importance of bacterial originated laccase in lignin degradation. There were 12 OTUs directly linked to xylanase, cellulose, and hemicellulose simultaneously, 11 of them were BOTUs but only one was FOUT, and three *Limnochordaceae* BOTUs and two *Savagea* BOTUs were contained. BOTU991 (*Savagea*) was even directly linked to xylanase, laccase, cellulose, hemicellulose, and lignin (Table 1). These indicated the potential core function in lignocellulose degradation provided by bacterial members, especially *Limnochordaceae* and *Savagea*. As previous research reported, *Limnochordaceae* was one of the main fungal groups in a maize waste compost, *Limnochordaceae* cooperated with *Anaerolineaceae*, *Heliobacteriaceae*, *Micromonosporaceae*, *Sphaerobacteraceae*, *Nocardiopsaceae*,

and *Vulgatibacteraceae*, recycled the carbohydrates and polypeptides of dead cells under high temperature situation and accelerated the humification of compost materials [51], while *Savagea* was always isolated from a pig manure compost. The isolated strains were reported 3 tylosin-and tetracycline-resistant, Gram-positive staining, aerobic, and motile rod-shaped, while the detailed metabolic function for material transformation were unreported yet [52]. In addition, our research indirectly confirmed its potential in degradation of organic polymers during IAPRs composting.

## 5. Conclusions

In this research, we systematically investigated the variation of microbial community composition, carbon source composition and relative enzyme activities during a composting of *Cinnamomum camphora* leaf IAPRs. The results supported that hemicellulose and cellulose were the main carbon sources for microbes, followed by lignin. The trade-off between *Firmicutes* and *Actinobacteria*, and *Eurotiomycetes* and *Sordariomycetes* were the main traits for the variation on bacterial and fungal community composition, respectively. The SEM and network analysis found that bacterial community composition, instead of fungal community, was more responsible for the carbon source transformation and degradation during composting, mainly by releasing bacterial originated xylanase and laccase. In addition, we found several bacterial members, such as *Limnochordaceae* and *Savagea*, played potential metabolic hub roles for the degradation of organic polymers. This research has taken the first step in exploring the carbon sources degradation related microbial dynamics during IAPRs composting, which offers valuable information to the improvement of IAPRs compost processes, facilitates the development of decomposer additives that targets to IAPRs compost, and provides new sights for the sustainable development of aromatic plant industries.

**Author Contributions:** Conceptualization, H.Z. and X.W.; methodology, H.Z.; software, H.Z.; validation, H.Z., L.D. and X.H.; formal analysis, H.Z.; investigation, H.Z. and X.W.; resources, X.W.; data curation, H.Z., T.D. and X.W.; writing—original draft preparation, H.Z.; writing—review and editing, H.Z.; visualization, H.Z., L.D. and X.H.; supervision, X.W.; project administration, X.W.; funding acquisition, X.W. All authors have read and agreed to the published version of the manuscript.

**Funding:** This research was funded by Jiangxi Science and Technology Department Major Science and Technology R&D Special Project, grant number 20203ABC28W016 and Jiangxi Forestry Bureau Camphor Research Special, grant number 2020-04-04.

**Institutional Review Board Statement:** Not applicable.

**Informed Consent Statement:** Not applicable.

**Data Availability Statement:** The raw sequences were uploaded to NCBI and the projects numbers are PRJNA914206 (fungal) and PRJNA914198 (bacterial).

**Conflicts of Interest:** The authors declare no conflict of interest.

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
