# Peer review of "Bacterial Community Drives the Carbon Source Degradation during the Composting of Cinnamomum camphora Leaf Industrial Extracted Residues"

_2036-7481, doi:10.3390/microbiolres14010019_

Round 1
Reviewer 1 Report
This work is very nice, but I have detected some mistakes. I hope that you find this review productive.
Abstract:
- Would be interesting to indicate why carbon degradation is more relevant than other elements in the residues, like N or S, to give strength to the work (later in the Introduction is someway better explained, but Abstract requires the anticipation of this) Introduction: - It's a good Introduction, but have some mistakes that make little difficult to follow and read it - Some connections between phrases are not very good; that is, Line 56 or 63 or 98 or 116... "And nearly"... Probably the first time I see a phrase beginning with 'And' Line 58... "Without proper...", this phrase is not connected, but also not well worked, is not conclusive or coherent with the rest of the text. Connection, flow and coherence between sections are needed here Line 127: Be careful with English use. "We hypotesis that..." Materials and Methods: - It is in general good; however, it is extremely short in many sections. I appreciate it help to read fast, but techniques descriptions and minor changes to them to adapt to your specific protocol are required to help the reader and to fully discern the utility of the methodology - Line 139 or 144 or 145...: Careful, but temperature degrees and percentages, rest of the units have a space in respect to the values Line 148: Measure with which device? And, hydrated in the case or how? Line 150: Device for pH? Results: - Line 199: conductivity is not expressed in % - Line 201: This phrase is not necessary - Be careful, the correlations do not imply causal reasons. If so, would require more tests to ensure it - In general, correlation results are not linked to an experimental testing of them, not necessary a big issue if not the purpose of this work, but some of the conclusions and reported results are not that string as they claim for without those tests Discussion: - Very cool. Just be careful with the strength of the correlation analyzeAuthor Response
Dear reviewer1:
Happy new year and best wishes to you, thanks for your kindness and constructive suggestions and corrections, and I’ll respond to them one by one in the following parts.
Comments and Suggestions for Authors
This work is very nice, but I have detected some mistakes. I hope that you find this review productive.
Abstract: Would be interesting to indicate why carbon degradation is more relevant than other elements in the residues, like N or S, to give strength to the work (later in the Introduction is someway better explained, but Abstract requires the anticipation of this)
Response: We added “Carbon source degradation is a basic but important field of compost research,” in the Abstract
Response: We added “Carbon source decomposition is an area of great interest in composting research, which is directly related to the quality of the compost products and the release of greenhouse gases, such as CO2 and CH4 during the composting process” in the Introduction.
Introduction:- It's a good Introduction, but have some mistakes that make little difficult to follow and read it - Some connections between phrases are not very good; that is,Line 56 or 63 or 98 or 116... "And nearly"... Probably the first time I see a phrase beginning with 'And' Line 58... "Without proper...", this phrase is not connected, but also not well worked, is not conclusive or coherent with the rest of the text. Connection, flow and coherence between sections are needed here Line 127: Be careful with English use. "We hypotesis that..."
Response: We deleted the “And” in line 56, the “Without proper...” in line 63 and “However” in line 98 and “Thus,” in line 116
Response: We added “Considering the generally stronger capability of fungi in decomposing recalcitrant organic carbons in the plant material composts” as a flow in line 127.
Response: We changed the phrase as “hypothesize”
Materials and Methods: - It is in general good; however, it is extremely short in many sections. I appreciate it help to read fast, but techniques descriptions and minor changes to them to adapt to your specific protocol are required to help the reader and to fully discern the utility of the methodology- Line 139 or 144 or 145...: Careful, but temperature degrees and percentages, rest of the units have a space in respect to the values Line 148: Measure with which device? And, hydrated in the case or how? Line 150: Device for pH?
Response: We added “by daily measurement and artificial watering” accordingly.
Response: We checked the unit format for the whole ms
Response: We added “a pH meter (FE20-FiveEasyTM pH, MettlerToledo, Germany)” accordingly.
Results: This phrase is not necessary - Be careful, the correlations do not imply causal reasons. If so, would require more tests to ensure it - In general, correlation results are not linked to an experimental testing of them, not necessary a big issue if not the purpose of this work, but some of the conclusions and reported results are not that string as they claim for without those tests
Response: We deleted several unnecessary causal reason descripts, including line 209, line 220, and we’ll use correlation causal analysis more carefully in the future studies, thanks for the valuable suggestions.
Discussion:
Very cool. Just be careful with the strength of the correlation analyze
Response: We changed several phrases accordingly, and thanks for the reminds.
Lastly, we sincerely thank the helpful corrections and suggestions offered again and best wishes for you.

Reviewer 2 Report
the article named Bacterial community drives the carbon source degradation during the composting of Cinnamomum camphora leaf industrial extracted residues presents useful information on composting of vegetal material. The paper is well presented ( sophisticated Figures); however, it is hard to read, thus, it can be improved.
Moreover, authors can fix/improve the following:
Lines:
80: types of microbes
83: and thus,...
127: We hypothezise (not we hypothesis
209:to improve:
212: decrement or decrease?
with coefficients...
line 246: A stage or at stage A
lines 279: the subfigures ? (to check and use Figure...
Fig 4: to check for: soluable?- sugars
421: has found to act as ...
426:that are transformed...
Author Response
Dear reviewer2:
Happy new year and best wishes to you, thanks for your kindness and constructive suggestions and corrections, and I’ll respond to them one by one in the following parts.
Comments and Suggestions for Authors
the article named Bacterial community drives the carbon source degradation during the composting of Cinnamomum camphora leaf industrial extracted residues presents useful information on composting of vegetal material. The paper is well presented (sophisticated Figures); however, it is hard to read, thus, it can be improved.
Moreover, authors can fix/improve the following:
Lines 80: types of microbes
Response: We had changed the term with “certain species of active microbes”
Lines 83: and thus
Response: We had deleted “thus”
Lines 127: We hypothezise (not we hypothesis
Response: We had corrected it as “hypothesize”
Lines 209:to improve
Response: We had deleted “which indicated that they were the main energy and carbon source for biosynthesis, especially hemicellulose” to make the phrase more scientific.
Lines 212: decrement or decrease?
Response: We had changed “decrement” to “reduction” to avoid misunderstanding
Lines 209: with coefficients
Response: We corrected it as “and the correlation coefficients were respectively -0.594 and -0.559”
line 246: A stage or at stage A
Response: We corrected it as “stage A”
lines 279: the subfigures ? (to check and use Figure...
Response: We changed the phrase as figure 2 C, and check for whole manuscript
Fig 4: to check for: soluable?- sugars
Response: We corrected as “Soluble”
Lines 421: has found to act as ...
Response: We changed the phrase as “were the main genera”
Lines 426: that are transformed
Response: We use “generated” instead
Lastly, we sincerely thank the helpful corrections and suggestions offered again.

Round 2
Reviewer 2 Report
Dear editor, the manuscript was improved accordingly to the proposed suggestions.